# Evolution and Driving Mechanism of Tourism Flow Networks in the Yangtze River Delta Urban Agglomeration Based on Social Network Analysis and Geographic Information System: A Double-Network Perspective

**Yuewei Wang [1,\*], Mengmeng Xi [1], Hang Chen [2,\*] and Cong Lu [1]**

[1] School of Business, Liaoning University, Shenyang 110036, China; xi18802419610@163.com (M.X.); 15254176020@163.com (C.L.)
[2] School of Tourism Management, Shenyang Normal University, Shenyang 110034, China
\* Correspondence: yueweiwang@lnu.edu.cn (Y.W.); chenhang-119@163.com (H.C.)

**Abstract:** This study's purpose was to analyze the network structural characteristics, nodal roles, spatial structure, and evolution laws from the dual network perspective, and apply the Quadratic Assignment Procedure (QAP) to conduct correlation analysis and regression analysis on the influencing factors of tourism flow networks in the Yangtze River Delta urban agglomerations. Using a mixed-method of social network and spatial analysis, Ucinet and ArcGIS software were used to comprehensively analyze the nodes in the travel routes covered in travel notes. The results show that the density of tourism flow network increases on the whole, while the spatial difference decreases, and the overall network density value is much lower than the average of the network density of provinces. Degree centrality, closeness centrality, betweenness centrality and the core–periphery structure analysis were used to examine the tourism function, distribution function, connection function and the position of nodes in the network, and nodes were divided into various types of roles according to their function. Meanwhile, the role changes of each node in different periods were also investigated. This study also builds an evaluation model of the influencing factors of the evolution of tourism flow network structure and uses QAP to find that the tourism network is affected by factors such as tourism resource endowment, transportation convenience, economic development level, tourism reception and service capacity. The research results are helpful for the Chinese government and tourism enterprises to understand the spatial behavior of tourists and its evolution rules, and to clarify the role and status of node cities in the tourism flow network and their influencing factors. It is of great significance for the formulation of joint marketing measures and promotion of the sustainable development of tourism in the Yangtze River Delta urban agglomeration.

**Keywords:** tourism flow network; SNA; GIS; Yangtze River Delta

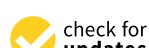



## 1. Introduction

Tourism flow is a phenomenon of the collective spatial movement of tourists in a large or small region caused by similarity of tourism demand. In the 1960s, many scholars discussed the spatial model of tourism flow and its impact. The earliest tourism flow spatial structure model is based on the theory of the "origin of trip-destination system", that is, the typical tourism flow model between two places (origin of trip-path-destination), which emphasizes the role of the path in the spatial structure [1]. Stewart and Lue put forward the spatial model of multi-destination tourism flow, emphasizing that tourism flow spreads through nodes of different levels or specific traffic systems. This model focuses on the flow characteristics of tourists among destinations of different levels and scales, and the study of the multi-destination interaction model contains the idea of tourism flow network [2]. Scholars have carried out research on different types of tourism flows, which help in

understanding the behavior of tourism flows, reveal its laws, explore its spatial impact, and optimize the spatial structure of tourist destinations [3–5]. Leiper (1979) regarded the five elements of the tourism system as: tourists, a tourism industry, origin regions, transit routes, and destination regions [6]. Taking these five aspects into consideration, tourism flow refers to the arrival of tourists from a place of origin to a destination through transit areas, and the stay of tourists in these areas [7]. Therefore, the actual travel routes of tourists can be used to define tourism flows. Furthermore, this study considers the tourism flow network as a network composed of tourist destinations and tourism flows among them. The tourists' multi-destination travel behavior based on their diversity preference, high grade preference, and pursuit of low unit cost makes the interaction of destination space increasingly frequent and multi-directional, resulting in a more complex tourism flow network [8].

As for the network characteristics of the tourism phenomenon, scholars have used social network analysis to study tourism problems since the early 2000s, and the ideas and methods of social network analysis have been widely used in tourism research [9–11]. Social network theory makes a deeper analysis of the tourism phenomenon based on the idea of relationship structure. If tourist destinations are regarded as actors within the tourism flow network, and tourist routes as connections between destinations, there is bound to be some kind of social relationship between tourist destinations [10]. More and more scholars are willing to use the Social Network Analysis (SNA) methodology to measure such social relations among tourist destinations and describe their social network characteristics [11]. For example, Lozano and Gutierrez (2018) analyzed the structure and interaction between tourist sources and destinations in the global tourism network using the SNA method [12]. Shao et al. (2020) applied SNA to discuss the structural characteristics and evolution rules of international tourism flow networks, as well as the roles and functions of various destinations in international tourism flow networks [13]. Wang et al. (2021) believe that the spatial movement of tourists from source to destination, and between destinations, contributes to a regional tourism cooperation network, and they used SNA to analyze the structure of tourism scenic spot cooperation networks [3]. Further, Wang et al. (2022) made a comparative analysis of tourism cooperation networks by using SNA from the perspective of both supply and demand [8].

In addition to SNA, Geographic Information System (GIS) can also describe the geospatial network characteristics of a tourism flow network. Many scholars have studied tourism flows in different spatial scopes [14,15]. It is of great significance for us to understand the different forms and characteristics of tourism flows in different spatial scales [5]. A tourism flow network and tourist distribution can be visualized by using the GIS network element spatial structure analysis tool. In GIS, the spatial structure of network elements can be investigated, and many aspects of the network's performance characteristics can be analyzed and calculated, using the mathematical theoretical model according to the topological relationship of tourism flow networks (topology between nodes and arcs, connectivity of arcs) [16,17]. GIS has been widely used to analyze the spatial distribution and relationship of tourism flows [18,19]. For example, Wang (2016) applied the GIS analysis method to analyze the kernel density of tourism flows in Lanzhou, China, and explained its space-time characteristics [20]. Gao et al. (2021) analyzed the spatial differentiation and related characteristics of tourism flows within China by using Tencent migration big data [21]. Nistor (2021) used GIS to explore the spatial characteristics and differentiation rules of different types of tourism flows [22].

The literatures listed above can help us understand the two methods commonly used to study tourism flows at present. First, SNA can achieve accurate quantitative analysis of various relationships by measuring the basic properties, centrality, connectivity, and structural holes of the network, thus providing quantitative tools for the construction of a middle-level theory and a test of empirical propositions, and even for establishing a bridge between "macro and micro." For instance, this method is increasingly applied to multi-destination networks, where the core and periphery destinations can be divided by

calculating the core degree. Second, in the study of the spatial and temporal distribution and evolution of tourism flows, the GIS analysis method has a significant advantage. It can be used to visualize tourism flow data through the inverse distance weight interpolation and other methods, so as to have a more intuitive and in-depth insight into the relationship between destinations. GIS analysis tools have been widely used in the study of spatial patterns and characteristics of tourist flows [23]. More interestingly, more and more scholars tend to use big data to analyze tourism flows. Leung et al. (2012) analyzed patterns of tourism flow networks using UGC (User Generated Content) data from travel notes and social media data [10]. Yang et al. (2021) summarized the structure and characteristics of tourism flow networks in major tourist attractions in Nanjing, China [24]. The relevant studies of the above scholars prove that SNA and GIS analysis methods as well as big data can be effectively integrated into the study of tourism flows. However, scholars have not done enough research on online travel diaries to identify the movement patterns of travelers. In addition, an analysis of the tourism flow pattern based on either SNA or GIS analysis is not enough, especially the lack of the combination of the two methods.

The formation of a tourism flow network will be affected by many factors, and scholars' research focus has gradually shifted from the early destination environment, economic development level, facilities and components to social system and policy, technological innovation and other influencing factors. Medina et al., emphasized the seasonal effect of climate on tourist flow and revealed the impact of destination climate change on tourist flow [25]. Martín et al., analyzed the impact of the establishment of the train network in Spain in terms of increasing tourist flow and reducing seasonality, and proved that transportation conditions are an important factor determining the attractiveness of destinations and the scale of tourist flow, which can promote the sustainable development of local tourism [26]. Tourism flows will also promote the economic development of destinations, provide support for the construction and renovation of infrastructure and the promotion of tourism image of destinations, which in turn will attract more tourism flow. Shao et al., analyzed the influencing factors of international tourism flow and found that safety concerns, transportation accessibility or small populations had a strong impact on inbound tourism flow in Africa [13]. In addition, other factors such as the global and national economic situation and public health events represented by COVID-19 will also have an impact on tourism flows [13,27]. However, some scholars have found that allowing a greater flow of tourists will lead to over-tourism and environmental degradation in destinations, thus hindering the overall sustainability of tourism [28,29]. The sustainability of government policies must be given top priority to achieve responsible tourism and sustainable tourism in order to avoid the massive increase of tourist accommodation units, deterioration of environmental conditions and inappropriate behavior of tourists and local stakeholders brought about by over-tourism [29]. This is rooted in the spatial formation and complex interaction of tourism flow network.

Although the study of tourism flow network has attracted the attention of some scholars, it is still relatively rare to use both SNA and GIS methods to study it. Moreover, there is little research on the tourism flow network in the Yangtze River Delta urban agglomerations concurrently using both the above two methods, especially of the evolution and driving mechanism of the tourism flow network. In order to fill this research gap, this study uses the online travel diaries of related websites to conduct dual network analysis of tourism flows. This study aims to analyze the network characteristics, spatial structure, and evolution laws from the dual network perspective, and apply the Quadratic Assignment Procedure (QAP) to conduct correlation analysis and regression analysis on the influencing factors of tourism flow networks in the Yangtze River Delta urban agglomerations. Consequently, the research questions of this study are as follows: What is the network structure and density of tourist flow? Is there any difference in the network structure and density of tourist flow in different periods, and what is the trend? What roles do different cities play in the tourism flow network? What is its spatial distribution law? Is there a spatial agglomeration effect and a law of spatio-temporal evolution? What is

the driving mechanism for the formation of a tourism flow network? These questions also reflect views on the density, characteristics of nodes, spatial structure, role and driving mechanism for the formation of tourism flow networks, which can provide valuable insights for the government and tourism enterprises to formulate appropriate policies and measures for regional tourism cooperation, planning tourism, designing tourism routes, and developing regional joint marketing. To answer these questions, five sub-objectives are established.

First, this study selects network size, network density and core-periphery structure to evaluate the social network structure of tourism flows in the Yangtze River Delta urban agglomeration.

Second, this study selects degree centrality, closeness centrality and betweenness centrality to evaluate node functions in the constructed tourism flow network. SNA method is used in the two aspects above, which can better reflect the relationship between nodes of tourism flow network.

Third, this study further analyzes the spatial characteristics and evolution trend of node indicators (e.g., network density and centrality) of tourism flow network by using a GIS analytical method, which can better reflect the physical state of the tourism flow network.

Fourth, this study investigates the factors in tourism flow by using QAP analysis, which can affect the formation of the tourism flow network.

Fifth, based on the travel diaries posted by tourists on the Internet, this study provides a new concept and basis for travel agencies to make a reasonable itinerary, and scientifically determine the most important cities and transit stations on the route according to the needs of tourists.

## 2. Materials and Methods

### 2.1. Study Area

According to the Yangtze River Delta Urban Agglomeration Development Plan issued by The State Council of the People's Republic of China in 2016, this study selected the following 26 cities as its research areas: Shanghai; Hangzhou, Ningbo, Jiaxing, Huzhou, Shaoxing, Jinhua, Zhoushan and Taizhou in Zhejiang Province; Nanjing, Wuxi, Changzhou, Suzhou, Nantong, Yancheng, Yangzhou, Zhenjiang, Taizhou in Jiangsu Province; and Hefei, Wuhu, Ma'anshan, Tongling, Anqing, Chuzhou, Chizhou and Xuancheng in Anhui Province (Figure 1). These 26 cities cover an area of 222,000 km², accounting for about 2.2% of China's total area. The urban agglomeration in the Yangtze River Delta has a vast economic hinterland and is an important engine for the economic development of the whole of China. The land, sea, and air transportation system is perfect, and tourism flows between regions close by. The Yangtze River Delta with its rich tourism resources, superior location conditions, and complete facilities, is one of the regions in China with better tourism development. Thus, it is representative to select the urban agglomeration in the Yangtze River Delta as a case study for tourism flow.

### 2.2. Research Framework for Analyzing the Tourism Flow Networks

In this study, we designed a new research framework, as shown in Figure 2, to analyze the social relations, spatial pattern and evolution among nodes of the tourism flow network and its influencing factors.

This research framework takes online travel diary text as data source, and uses Python web crawler to retrieve relevant web pages and realize data collection and processing. The dual network perspectives referred to in this study are social network perspective and spatial network perspective, respectively. First, from the perspective of social network, we construct three social relationship networks of tourism flows in the Yangtze River Delta urban agglomerations in different periods, evaluate their network structure, network node function and interaction relationship, respectively, and summarize their differences. This part is also the core of the research framework. Second, from the perspective of spatial

network, we construct three geospatial networks of tourism flows in the Yangtze River Delta in different periods based on network density and node centrality in the social network of tourism flows. In order to observe the spatial pattern and evolution of tourism flow network more directly, we use the inverse distance weight interpolation analysis of the GIS analysis method to express the spatial visualization of the tourism flow social relationship network index, and analyze its evolution trend. Third, the factors affecting the formation of tourism flow in the Yangtze River Delta urban agglomeration are found through QAP analysis. These sections are discussed in more detail below.

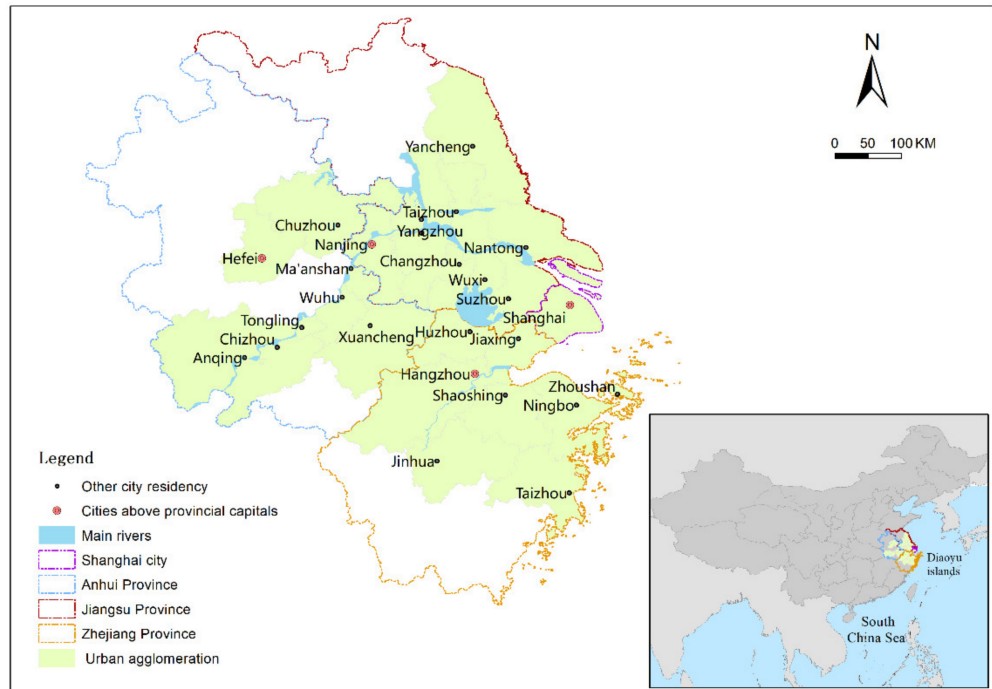

**Figure 1.** The location of the study area (Yangtze River Delta, China).

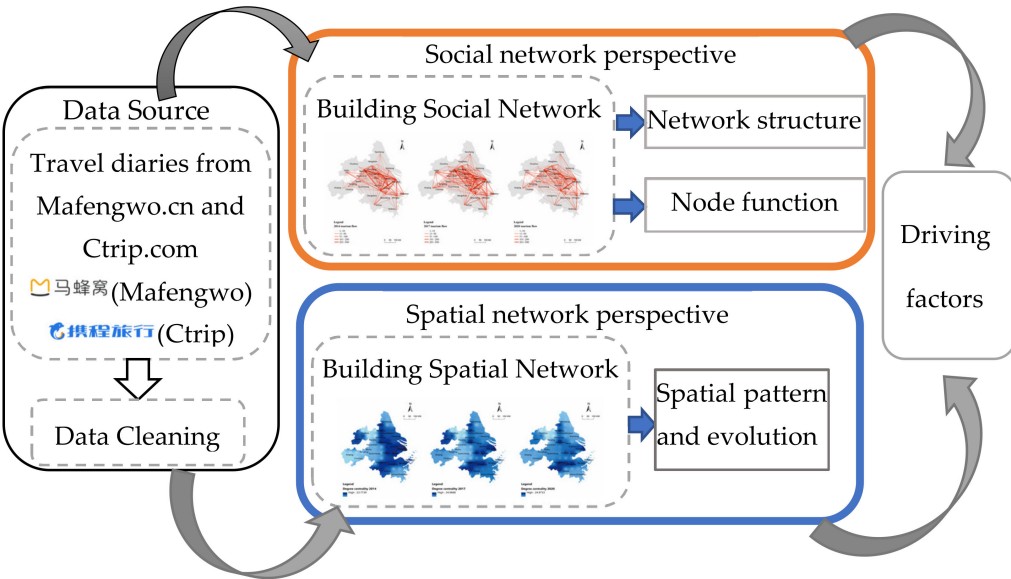

**Figure 2.** Research framework for analyzing the tourism flow networks.

### 2.3. Data Collection and Analysis

Previous research data on the spatial behavior characteristics of tourists mainly come from questionnaire surveys, internal materials of tourism enterprises, tourism newspapers, annual reports of professional tourism market research institutions, etc. [16–19], which have certain limitations in reflecting tourists' spatial behavior characteristics. The rapid development of the Internet provides the possibility of in-depth mining of tourism data. Among these, the travel diaries of virtual communities voluntarily written by tourists, are relatively authentic in expressing tourists' subjective behaviors.

First of all, the Python crawler was used to crawl the travel diaries with complete travel routes published by tourists in 2014, 2017 and 2020 from the websites of Mafengwo (www.mafengwo.cn) and Ctrip (www.Ctrip.com), with a total of 23,522 travel notes on 1 May 2021. Data cleaning is then performed. The authenticity of the travel notes was confirmed by consulting the locations, photos, air tickets, hotel accommodation and other information in the travel notes. By excluding some business travel notes of interest and including at least two cities in the Yangtze River Delta urban agglomeration, 7963 valid travel notes were finally obtained. Finally, the travel routes in travel notes are summarized and sorted out. Mafengwo (http://www.mafengwo.cn) having hundreds of millions of users' authentic travel experience texts, including travel notes, guides, notes, and comments, is the most popular travel social network among China's young generation. Ctrip (http://www.ctrip.com), providing accommodation booking, transportation ticketing, travel diary sharing, and other services, is China's largest one-stop travel platform. These websites record the travel diaries voluntarily written by tourists on their travel experiences in different destination cities in the Yangtze River Delta region, and as they visit more than one destination in a single trip, their itineraries also include more than one destination.

In this study, we apply social network analysis to explore the social relations between nodes (cities) of the tourism flow network, in which nodes (cities) are regarded as "actors" and tourist routes between cities are regarded as "links". For example, Figure 3a is a network diagram showing tourist travel routes in five cities, from which an asymmetric matrix can be constructed (Figure 3b). According to the tourism routes obtained after data cleaning, the original matrix table of 26 rows × 26 columns is established by such deduction and accumulation. The starting point of behavior is listed as the end point, and the original matrix of tourism flows in 2014, 2017 and 2020 is finaly obtained.

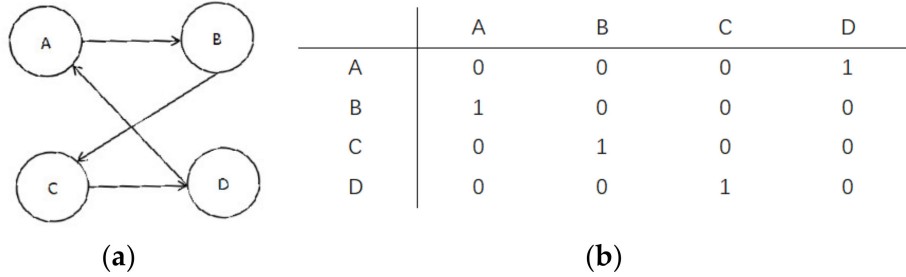

(**a**)                     (**b**)

**Figure 3.** A simple case with four actors. (**a**) Network diagram; (**b**) Network matrix.

### 2.4. Research Methods

Although SNA, GIS and QAP methods have their own unique advantages in tourism flow network research, it is still very rare to systematically use these three methods to study the spatial structure of tourism flow network and the role of network nodes, especially the spatio-temporal differentiation of tourism flow and its driving factors [30]. None of these methods can either fully evaluate the density of a tourism flow network, the relationship, status, and connection mode among its destinations from the perspective of a dual network, or further reveal its law of spatio-temporal differentiation and driving factors. Moreover, no research has been found on the tourism flow network of the Yangtze River Delta urban agglomeration using the three methods simultaneously. The combination of SNA, GIS and QAP analysis methods mainly based on sociology and graph theory can measure

and plot the patterns, flow, intensity and influencing factors of the relationship between destinations, making tourism flow network analysis different from other analysis methods when analyzed from the perspective of a double network.

The tourism flow of the Yangtze River Delta urban agglomeration was studied by referring to the existing research results [3,8,10,11,22], using SNA and GIS methods, and the QAP. In network analysis, the cohesion of the tourist destination and its network structure characteristics are usually tested by density and centrality. Density is the ratio of the number of relationships that exist between destination nodes to the total number of possible associations if each member is bound to another. Usually, density is calculated as a fraction of the actual number of links between tourist destinations as the maximum number of possible connections. It is important to consider density because highly dense networks have a stronger exchange of information, ensuring the flow of institutional norms and generating common behavioral expectations. Density is a characteristic of the entire network and measures the relative number of associations that tie participants (tourist destinations) together [31]. Centrality refers to the power acquired by actors (tourist destinations) through structures rather than individual attributes [31]. Core-periphery structural analysis is designed to study which nodes are at the Core and which are at the Periphery of the social network. Core-edge structure analysis has a wide range of applications, which can be used to analyze the core-edge structure in various social phenomena such as an elite network, scientific citation network and organizational relationship network.

This study's content includes three aspects: network structure evaluation, nodal function evaluation, and influence factor analysis. Among these, network size, network density, and core–periphery structure were selected as indicators for network structure evaluation; degree centrality, closeness centrality, and betweenness centrality were selected as indicators for nodal function evaluation; QAP regression analysis was used for influencing factors analysis; and ArcGIS software was used for data visualization. The calculation method and interpretation of some of these indicators are as follows.

### 2.4.1. Network Density

This index represents the actual number of relationships of nodes, divided by the theoretical maximum number of relationships. The greater the overall network density, the closer are the connections among the network members reflected [3]. The standard formula for computing the network density is:

$$D = \frac{2 \sum_{i=1}^{m} d_i(i)}{m \times (m-1)} \qquad (1)$$

$$d_i(i) = \sum_{j=1}^{m} d_i(d_i, d_j) \qquad (2)$$

In Formula (1), $D$ is network density; $m$ is the total number of cities (destinations) in the Yangtze River Delta. In Formula (2), $d_i(d_i, d_j)$ serve as relational functions; when $i, j$ have a direct link, both $d_i(d_i, d_j) = 1$ on the contrary, $d_i(d_i, d_j) = 0$.

### 2.4.2. Core-Periphery Model Analysis

The core-periphery model is used to reflect the position of each city (destination) in the tourism flow network, reveal which cities (destinations) are in the core and periphery regions, and find the inner relationship between the core and periphery regions. SNA provides many analysis models, among which the core-periphery relationship missing model is suitable for analyzing the binary directed data in this study. The network/Core and Periphery/Categorical analysis module of UCINET software can be used to analyze the tourism flow relationship matrix. In addition, the core of each city (destination) can be measured using a continuous core-periphery association model (network/Core and Periphery/Continual in UCINET) [8].

### 2.4.3. Node Centrality

Node centrality is mainly used to reflect the importance of nodes, including degree centrality, closeness centrality, and betweenness centrality.

Degree centrality is a basic indicator to describe the function of network nodes. The degree of node *i* represents the sum of the number of nodes connected to it, and is a centralized reflection of the importance and influence of nodes. If a node is directly related to multiple nodes in the network, then the node is in the center of the network [3,8]. The calculation Formula (3) of degree centrality is as follows:

$$C_D(i) = \sum_{j=1}^{n} x_{ij} \tag{3}$$

$C_D(i)$ represents the degree centrality of node *i*; $x_{ij}$ is the tourism flow between *i* and *j*.

Betweenness centrality measures the ability of a node to act as an intermediary in the whole network. If a node is on many other "node-pair" shortcuts (i.e., the shortest path between two nodes), the node has a high betweenness centrality. Its calculation Formula (4) is as follows:

$$C_B(i) = \sum_{j=1}^{n} \sum_{k>j}^{n} \frac{g_{jk}(i)}{g_{jk}} \tag{4}$$

$C_B(i)$ represents the betweenness centrality of node *i*; $g_{jk}$ is the number of the shortest path between *j* and *k*; $g_{jk}(i)/g_{jk}$ represents the probability that *i* is on the shortest path between *j* and *k*; *n* is the number of nodes associated with tourism flow with node *i*.

Closeness centrality is calculated by measuring the number of lines contained in the shortest path between an actor and other actors in the network. The closeness centrality is negatively correlated with the distance between each point. If this value is larger, it means that the node is closer to other nodes. The core point needs to reach this point through other nodes, and has strong dependence on other nodes. Its calculation Formula (5) is as follows:

$$C_{Ci} = \frac{1}{\sum_{j=1}^{L} d_{ij}} \tag{5}$$

*L* is the number of other nodes with high traffic accessibility to node *i*. $d_{ij}$ is the traffic distance between nodes.

### 2.4.4. QAP Regression Analysis

QAP is a quantitative analysis of the relationships between matrices. Given that, tourism flow is a kind of network relational data, using the traditional regression analysis method for the study of its influencing factors encounters the data multicollinearity problem, whereas the QAP method has been proved to be better able to avoid this problem [32,33].

The basic principle of QAP regression analysis is to compare each element in the two matrices, calculate the correlation coefficient between them, and conduct a non-parametric test on the coefficient. When the significance level is within 0.01, 0.05, or 0.10, it indicates that the two matrices have a strong correlation in the statistical sense. At present, this method has been common in tourism, sociology, and geography research [32,34–36]. Based on the existing research results of relevant scholars, this study constructed the evaluation model of influencing factors on the evolution of tourism flow network structure as follows:

$$R = f(PS, TRE, EDL, TRSA, TCD) \tag{6}$$

In Formula (6), *R* represents the tourism flow network relationship matrix; *PS* represents the difference matrix of policy support; *TRE* stands for the difference matrix of tourism resource endowment; *EDL* stands for the economic development level difference

matrix; *TRSA* stands for the difference matrix of tourism reception service capacity; *TCD* represents the matrix of traffic convenience degree.

(1) Policy support (*PS*): The development of tourism is dominated by the government. The government's implementation of active policies is conducive to accelerating the expansion of the tourism market, promoting complementary advantages of resources and balancing regional development. Therefore, we argue that the government's formulation and implementation of policies to promote the development of the tourism industry will increase tourist flows to destinations.

(2) Tourism resource endowment (*TRE*): Tourism resources can stimulate tourists' travel motivation and travel activities. Because of the imbalanced distribution of tourism resources in different destinations and their immobility, tourism flow comes into being. Therefore, we argue that unique tourism resources have a strong attraction, which can help tourists effectively overcome the resistance of spatial distance to a certain extent and form tourism flows.

(3) Economic development level (*EDL*): The level of economic development of tourist destinations plays an important role in the development of tourist resources, the construction of infrastructure and the improvement of the tourist environment. The higher the level of economic development of tourist destinations, the more investment in tourism development, the more perfect the construction of tourist attractions and infrastructure, the more tourists' needs can be met. Therefore, we argue that the level of economic development of tourist destinations is also an important factor influencing the formation of tourist flows.

(4) Tourism reception service capacity (*TRSC*): Tourism reception service is the most effective way to show the cultural connotation of a tourism destination, promote local culture and enhance attraction. Tourism reception service plays a very important role in shaping a good market image of a destination and improving comprehensive competitiveness of the tourism industry. Only with a high level of tourism reception service ability can the trust and reputation of tourists be won. Therefore, we argue that the tourism reception service capacity of a destination will affect the formation of tourism flows.

(5) Traffic convenience degree (*TCD*): Tourism transportation facilities and tools build bridges for tourists to and from the source and destination, and are the basic conditions for improving the accessibility of a destination and promoting the sustainable development of the tourism market. Whether the traffic is convenient or not will directly affect the formation of tourism flow between two cities. Therefore, we argue that the formation of tourism flow is inseparable from the influence of transportation convenience.

## 3. Results and Discussion

### 3.1. Evolution Analysis of Network Density

Ucinet6.0 software was used to construct the network structure of tourism flows in 2014, 2017, and 2020, and ArcGIS 10.5 software was used to analyze the spatial visualization of tourism flows, so as to reveal the regional spatial distribution of tourism flows (Figure 4). In 2014, 2017, and 2020, the scale of the tourism flow network (Table 1) showed an overall growth trend. In 2020, owing to the impact of COVID-19, the scale of the network decreased slightly, compared with that in 2017.

**Table 1.** Tourism flow network density in 2014, 2017, and 2020.

| Year | Overall Network Density | Anhui Province | Jiangsu Province | Zhejiang Province |
|------|------------------------|----------------|------------------|-------------------|
| 2014 | 0.357 | 0.339 | 0.792 | 0.875 |
| 2017 | 0.572 | 0.661 | 0.847 | 0.929 |
| 2020 | 0.497 | 0.696 | 0.861 | 0.821 |

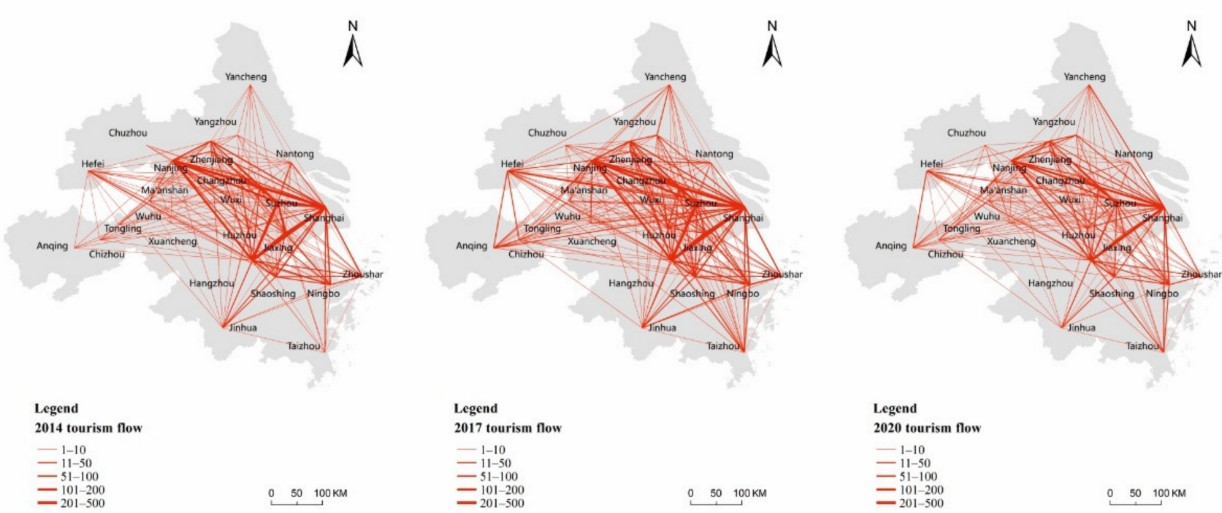

**Figure 4.** Spatial structure of tourism flow network in 2014, 2017 and 2020.

The value of network density directly reflects the development degree of tourism flow network. The closer the density is to 1, the closer is the connection of nodes in the network. The density of tourism flow network shows an increasing trend on the whole, indicating that its development is becoming more and more perfect, and the inter-city tourism flow is getting more and more closely connected. In 2020, for a period of time, owing to the impact of COVID-19, tourists were prevented from traveling, resulting in a slight decrease in the tourism flow network density. The spatial difference of tourism flow network density shows a decreasing trend. The maximum difference of tourism flow network density in Anhui, Jiangsu, and Zhejiang decreased from 0.536 in 2014, to 0.268 in 2017, and then to 0.165 in 2020. In addition, the overall network density value is much lower than the average of the network density of provinces, indicating that, influenced by administrative boundaries, the tourism flow within the province is significantly greater than that between provinces. Therefore, how to strengthen the inter-provincial tourism flow is the key to further improving the tourism flow network. On the whole, the density of tourism flow network of urban agglomerations in the Yangtze River Delta gradually develops from an unbalanced pattern of tight northeast and loose southeast to a relatively balanced pattern, the distribution pattern with Shanghai, Hangzhou, Suzhou and Nanjing as four cores evolves to a multi-core pattern, and the connectivity of urban agglomerations increases day by day. Although tourism resource distribution is relatively concentrated in Shanghai, Hangzhou, Suzhou, and Nanjing, these cities' tourism industry and economy, population, and transportation guarantee that their tourist flow is relatively stable, but with other cities artificial class tourist attractions to speed up the development and the improvement of the reception facilities, in particular, the interchange of high-speed railways and expressways between cities in the Yangtze River Delta, also promote the balanced evolution of tourism flow.

### 3.2. Evolution Analysis of Network Node Centrality

3.2.1. Degree Centrality Analysis

In 2014, 2017, and 2020, the degree centrality interval of cities were: (4.00009, 23.8573), (10.0001, 24.9998), and (8.0001, 25.0000), respectively (Figure 5). In 2020, in addition to the impact of the COVID-19 pandemic, the interval difference of degree centrality of cities showed a decreasing trend on the whole, indicating that the spatial distribution of degree centrality is gradually balanced, and the tourism flow network is evolving to multi-core, with a closer connection of tourism flows. Shanghai, Nanjing, Hangzhou, Suzhou and Yangzhou are the top five cities due to their superior location, convenient transportation and strong comprehensive competitiveness of their tourism, and they are in the "core"

position in the tourism flow network. The inverse distance weight interpolation analysis of degree centrality shows that, in 2014, the degree centrality was greater than 20 for five cities—Shanghai, Nanjing, Suzhou, Yangzhou and Hangzhou—located in the high value area and the "core" position in the tourism flow network. This may be related to the fact that these cities belong to municipalities directly under the central government or provincial capital cities or are tourist cities. These cities have good tourism resource endowment, high level of economic development, convenient transportation and perfect tourism supporting facilities, so they have a significant influence on the spatial network of tourism flow in the Yangtze River Delta urban agglomeration In 2017, three cities—Hefei, Wuxi and Huzhou—were added to the high-value zone. In 2020, Ningbo's degree centrality ranked among the high value areas, indicating that its tourism was developing rapidly, tourism industry policies were constantly improving, and attraction to tourists was increasing.

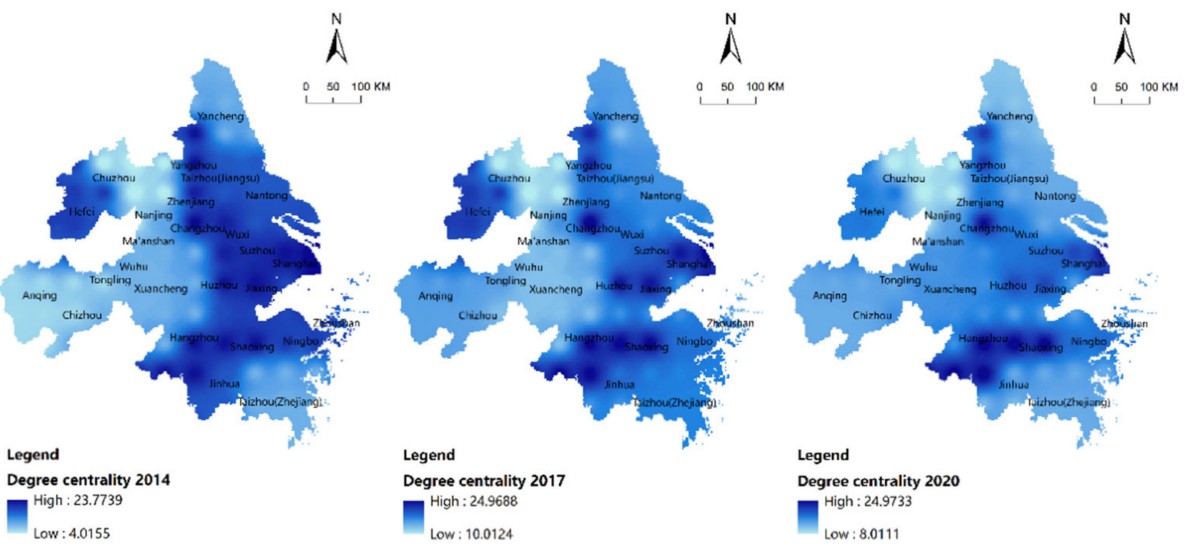

**Figure 5.** The degree centrality of the 26 cities.

### 3.2.2. Closeness Centrality Analysis

In 2014, 2017, and 2020, the closeness centrality interval of cities was (0.3113, 0.8705), (0.4729, 0.9999), and (0.4262, 0.9999), respectively (Figure 6). In 2020, apart from the impact of the COVID-19 pandemic, the difference between the closeness centrality showed a narrowing trend on the whole, indicating that the flow of tourists among cities became more frequent. Tourism flow network structure appears more point dependent. The inverse distance weight interpolation analysis was carried out for the closeness centrality of each city. The results show that in 2014, only Shanghai, Nanjing, Suzhou and Hangzhou were located in the high-value area and, having a greater "priority" in the regional tourism flow network, had a closeness centrality greater than 0.7. In 2017, the closeness centrality of each city had increased significantly. There were 19 cities whose closeness centrality was greater than 0.7, among which the closeness centrality of Shanghai, Nanjing, Hangzhou, Suzhou, Yangzhou, and Hefei were greater than 0.85. This shows that the independence of each city node in the regional tourism flow network is constantly improving. In 2020, owing to the impact of COVID-19, tourism flows in the region weakened, and the closeness centrality of nodes in each city decreased slightly, but there were still 12 cities with high closeness centrality in Hangzhou, Shanghai, Nanjing, Yangzhou, and Suzhou. Due to the development of high-speed rail, highways and other transportation industries, the core position of the above-mentioned cities has been strengthened, and their control over other surrounding node cities has been improved. However, Nantong, Changzhou, Zhenjiang, Anqing and other cities have a low tourism development level and strong dependence on

other tourism node cities due to the low visibility of their tourism resources and the poor perception of tourists.

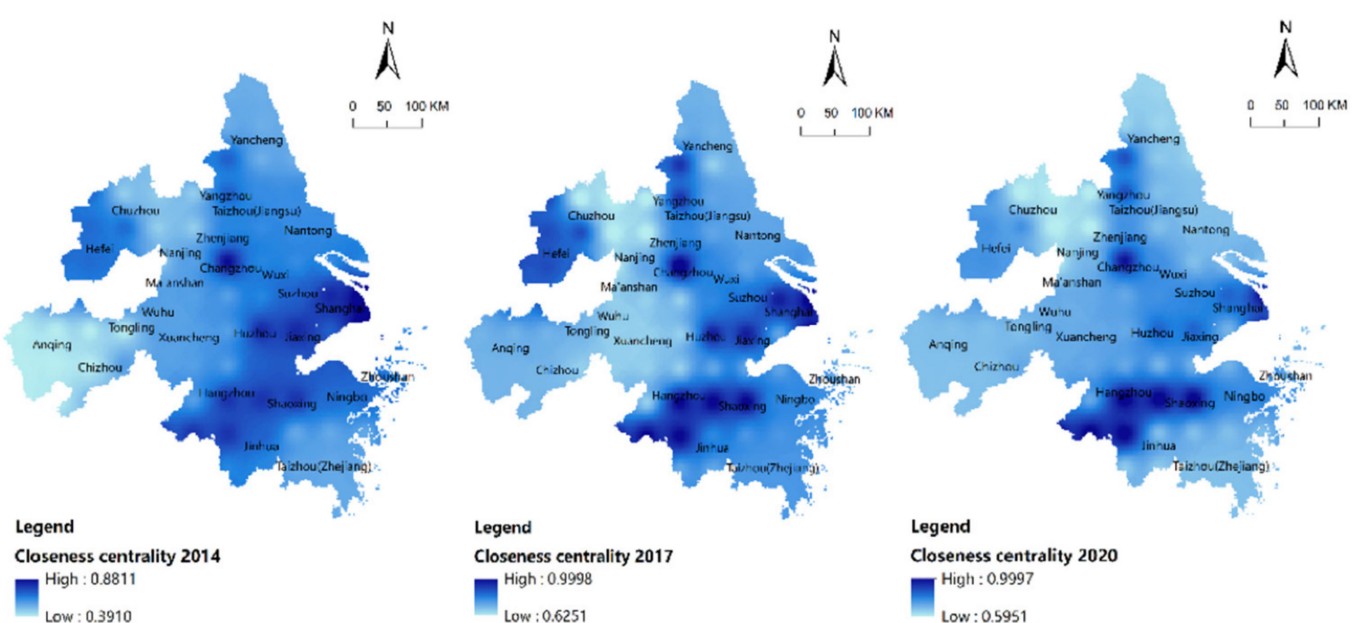

**Figure 6.** The closeness centrality of the 26 cities.

### 3.2.3. Betweenness Centrality

In 2014, 2017, and 2020, the betweenness centrality of each city was 2.77, 2.936, and 3.006, respectively (Figure 7), showing a gradually increasing trend, indicating that the power in the tourism flow network was becoming more and more concentrated, and the core node in the network had gradually strengthened its control over other nodes. The inverse distance weight interpolation analysis was carried out for the betweenness centrality of each city. The results showed that only Shanghai, Nanjing, Hangzhou, and Suzhou located in the high value region had betweenness centrality greater than 5 in 2014. In 2017, the number of node cities in the high value zone gradually increased. For example, Wuxi, Yangzhou, and Hefei entered the high value zone. In 2020, Jiaxing also entered the high-value zone. Affected by location, traffic, and other factors, nodal cities with high betweenness centrality, such as Shanghai, Nanjing, and Hangzhou, are playing an increasingly obvious role of "bridge" and "intermediary" in regional tourism flows. For some cities with poor location and traffic conditions, such as Anqing, Chuzhou, Taizhou (Zhejiang) and Tongling, Ma'anshan and Taizhou (Jiangsu), the betweenness centrality was always at a low value, and its mediating effect was almost zero. This is mainly due to the lack of tourism resources and low tourism visibility in these cities compared with well-developed cities such as Shanghai, Suzhou, Hangzhou and Nanjing. With the increase of tourism flows in the region, the betweenness centrality of some cities with good location and traffic conditions increased significantly. For example, the betweenness centrality of Hefei has increased from 0.881 in 2014, to 6.046 in 2017, and then to 10.752 in 2020. Although the betweenness centrality of Hefei is still low, it has high access in the tourism network and is an important tourism channel, with strong control over other node cities.

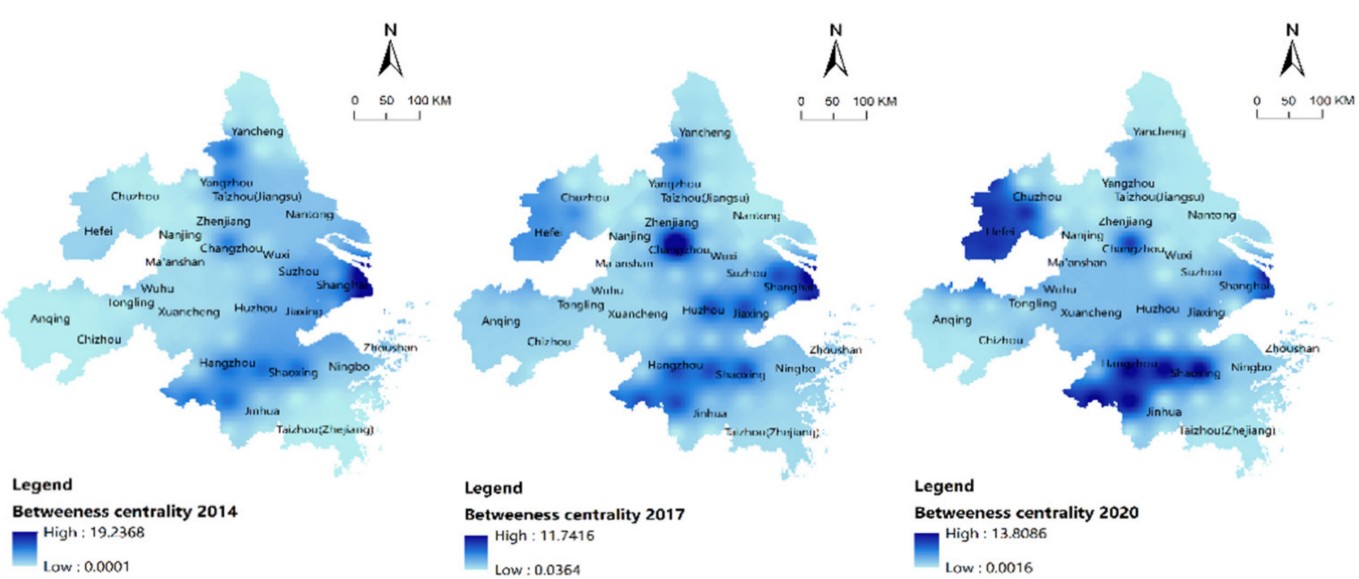

**Figure 7.** The betweenness centrality of the 26 cities.

3.2.4. The Core–Periphery Structure Analysis

In 2014, 2017, and 2020, the values of the core degree of each city node showed little change (Figure 8), indicating that the core-edge structure of the tourism flow network presents an overall stable situation. In addition, the core degree of the 26 cities tends to be balanced. Shanghai, Hangzhou, Suzhou, and Nanjing have always been in the high core degree area (>0.25), and the gap between them keeps narrowing. Cities in the low value area (<0.1), including Yancheng, Taizhou, Taizhou, Ma'anshan, Tongling, Anqing, Chuzhou, Chizhou, and Xuancheng, showed an obvious upward trend in core degree.

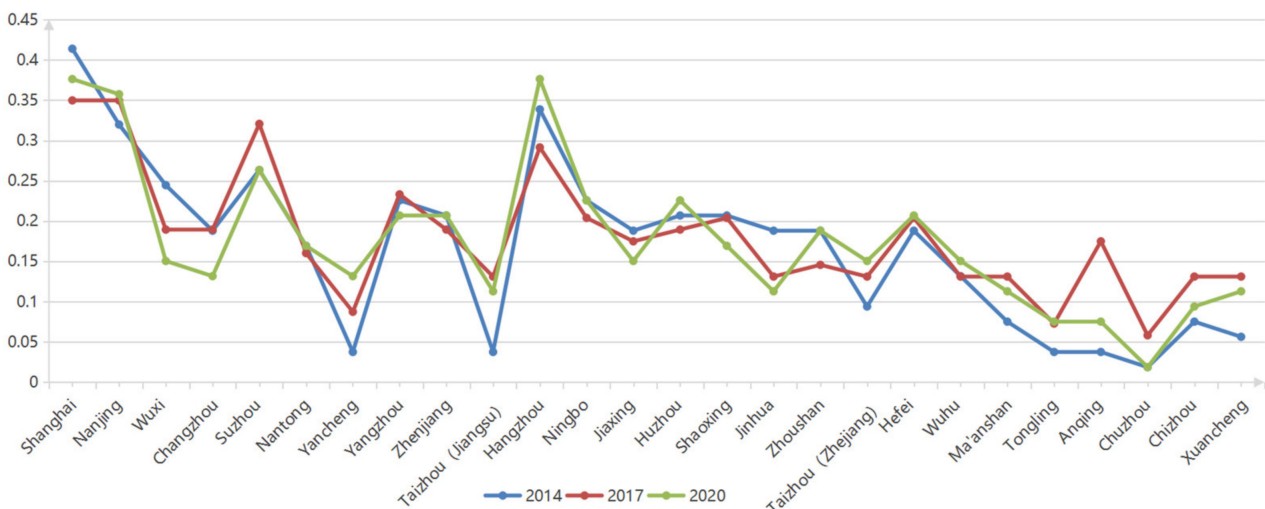

**Figure 8.** The core degree of tourism flow network in 2014, 2017 and 2020.

The network/core and periphery/categorical analysis module of UCINET software was used to analyze the tourism flow relationship matrix, and the spatial visualization of the analysis results was carried out using ArcGIS software (Figure 9). The contents of Figure 6 show that the number of cities in the core area decreased from 15 in 2014, to 12 in 2017, and then to 11 in 2020, which indicate that the core area of tourism flow presents a shrinking trend, mainly including Shanghai, Nanjing, Suzhou, Yangzhou, Wuxi, and Zhenjiang in Jiangsu Province; Hangzhou, Ningbo, and Huzhou in Zhejiang province; and

Hefei in Anhui Province. This shows that the agglomeration effect of regional tourism flow is extremely significant.

**Figure 9.** The core-periphery region of tourism flow network in 2014, 2017, and 2020.

### 3.3. Driving Mechanism Analysis of Tourism Flow Network Formation

3.3.1. The Selection of Drivers

The process of tourists choosing destinations is influenced by many factors, including their own subjective factors, such as gender, age, and motivation, as well as objective factors, such as destination tourism resources and transportation. This study mainly discusses the influence of destination objective factors on the formation of tourism flow networks. Current studies on influencing factors of tourism flow networks mainly focus on the reasons for the flow of tourists from their source place to destination [37–39], among which, the "thrust-pull-resistance" driving mechanisms, constructed from the tourism system theory perspective, is widely used [40]. However, previous studies have paid little attention to the factors affecting the flow of tourists in different destinations. For example, Dong et al. (2021) found that the spatial distribution of tourism flows are jointly affected by factors, such as tourism resource endowment, reception service capacity, and transportation convenience [41]. Li et al. (2020) found that tourism resource endowment, location and transportation, market demand, economic development, government behavior, infrastructure, and other factors have an impact on the spatial flow of tourism flows [42]. In their study, Mou et al. (2018) found that tourism reception capacity, tourism resource endowment, tourism destination visibility, and social security had a significant impact on the tourism flow network in Qingdao [43]. By studying the existing literature and combining the spatial flow rules of tourists in the Yangtze river delta urban agglomeration, tourism resource endowment, transportation convenience, economic development levels, policy support, and reception service capacity were selected as independent variables (Table 2) to study their influence on the formation of tourism flow networks.

To eliminate the influence of different dimensions, the range standardized processing of research data is carried out as follows:

$$X_{Std} = (X_i - X_{min})/(X_{max} - X_{min})$$

3.3.2. Driving Mechanism Analysis

Based on the correlation analysis of the tourism flow network and its influencing factor data matrix, the results show that tourism resource endowment, transportation convenience, economic development levels, and tourism reception service capacity are positively correlated with the tourism flow network. This is consistent with the research hypothesis, and further indicates that tourists tend to flow to cities with higher resource endowment, more developed economies, and perfect reception facilities (Table 3). Among

the variables of transportation convenience degree, the tourism flow network is negatively correlated with the variables "minimum travel time by automobile" and "minimum travel time by high-speed rail/train," but positively correlated with the variables "minimum travel time by air," which is inconsistent with the research hypothesis (Table 3). The main reason for this phenomenon is that, in short distance travel with a radius of less than 500 km, as the aircraft's speed advantage is not obvious, but the travel cost is relatively high, tourists have fewer choices. However, for long distance trips with a travel radius of more than 500 km, more tourists choose to travel by air. Thus, the longer the distance, the more passengers choose to travel by air. The shortest travel time of automobile, high-speed rail, or train can truly reflect the tourism flow, i.e., the shorter the minimum travel time, the stronger the relationship between the tourism flow, and the two are negatively correlated.

**Table 2.** The index system's meaning and data source.

| Driving Factors | Variable Names | Meaning and Data Source | Expectations | References |
|---|---|---|---|---|
| Tourism resources endowment | Number of 5A and 4A scenic spots | Tourists tend to flow to high-grade scenic spots. According to the collected travel notes and text data, the difference matrix is established by the sum of the number of 5A and 4A scenic spots. | + | [44,45] |
| Transportation convenience | Minimum travel time by automobile | Tourists are willing to move between easily accessible areas. The shortest time through Baidu map query. | - | [45,46] |
| | Minimum travel time by high-speed rail/train | Check China's Railway official website (https://www.12306.cn/) for a high-speed rail/train that takes the shortest running time. If there is no railway between the two places, the value assigned is 0. | - | [42,46] |
| | Minimum travel time by air | Inquire through the carrier network (www.ctrip.com) to get the shortest operating time of the aircraft. If there is no route between the two places, the value is 0 | - | [46,47] |
| Economic development level | GDP gross | Tourists tend to flow in areas with high economic levels. According to the statistical bulletins of cities in the Yangtze River Delta, the difference matrix is established by taking the values for 2020. | + | [45,47] |
| | Total income of tertiary industry | Ditto | + | [47] |
| | The proportion of tertiary industry in GDP | Ditto | + | [47] |
| Policy support | Tourism financial expenditure | The greater the government policy support, the more effective it will be to improve the destination supply and tourism attraction. According to the statistical bulletin of each city, the values for 2020 are taken to establish the difference matrix. | + | [48] |
| Tourism reception service capacity | Number of travel agencies and star-rated hotels | Tourists tend to flow in areas with strong reception and service capacity. The data were collected from the cultural and tourism departments of Anhui, Zhejiang, Jiangsu, and Shanghai, and the difference matrix was established according to the quantity. | + | [44,47] |
| | Number of tertiary industry employees | Ditto | + | [44,47] |

**Table 3.** The QAP analysis results of influencing factors of tourism flow.

| Item | Variable Names | QAP Correlation Coefficient | *p*-Value | QAP Regression Coefficient | Sig. |
|---|---|---|---|---|---|
| Tourism resources endowment | Number of 5A and 4A scenic spots | 0.298 *** | 0.001 | 0.157 ** | 0.013 |
| Transportation convenience | Minimum travel time by automobile | −0.228 *** | 0.000 | −0.195 *** | 0.001 |
| | Minimum travel time by high-speed rail/train | −0.264 *** | 0.000 | −0.121 ** | 0.003 |
| | Minimum travel time by air | 0.237 *** | 0.002 | 0.141 | 0.008 |
| Economic development level | GDP gross | 0.329 *** | 0.001 | 0.081 | 0.011 |
| | Total income of tertiary industry | 0.275 *** | 0.000 | 0.188 | 0.099 |
| | The proportion of tertiary industry in GDP | 0.249 *** | 0.000 | 0.401 | 0.001 |
| Policy support | Tourism financial expenditure | 0.204 *** | 0.358 | 0.006 | 0.107 |
| Tourism reception service ability | Number of travel agencies and star-rated hotels | 0.336 *** | 0.000 | 0.449 ** | 0.011 |
| | Number of tertiary industry employees | 0.296 *** | 0.002 | 0.508 ** | 0.014 |

Note: ** and *** passed the significance test at the significance level of 0.01, and 0.001, respectively.

UCINET software was used to conduct QAP regression analysis on the tourism flow network and influencing factor matrix. The random displacement times were set to 5000. The results showed that the significance probability was 0.000, and the adjusted $R^2$ was about 0.394. This shows that the difference of selected indicators can explain 39.4% of the spatial flow relationship of tourism flows in the Yangtze River Delta urban agglomeration. Previous research experience shows that the deterministic coefficient of the QAP model based on the same data is generally lower than that of the OLS (ordinary least squares) model [47]. In previous studies based on the QAP method, $R^2$ values were mostly between 12.5% and 40.3% [47,48]. This study's $R^2$ data is relatively moderate and the index fit is good. Therefore, it has good explanatory power.

The results of QAP regression analysis were basically consistent with the correlation analysis (Table 3), but some variables failed the significance test, which was similar to the conclusions of previous studies [44–47]. Among all the indicators that pass the significance test, tourism resource endowment has been proved to have a significant driving effect on the tourism flow, while the minimum travel time of automobile and high-speed rail/train has a significant hindrance effect. The three variables of the economic development level did not pass the test of significance, and did not agree with the previous research's conclusion [47], although the network analysis of the present study has demonstrated that the tourism flow of economically developed cities such as Shanghai and Nanjing have stronger gravitational pull, but this kind of phenomenon exists only in individual cities. From the perspective of the spatial flow of tourists, whether economically developed cities are more attractive or not, there are significant differences in the conclusions of studies conducted at different scales, which is also worthy of further discussion in the future. The tourism financial expenditure in government policy support also fails the significance test, which may be because not all of it is used to develop and utilize tourism products, or improve facilities, and other aspects that can attract tourists quickly in a short time. In addition, through initiatives such as formulating tourism plans, building common brands, and improving the tourism environment, the attraction effect on tourism flows can be reflected over a long period of time. In terms of tourism reception service capacity, the number of hotels, travel agencies, and tertiary industry employees all pass the significance test, which has

certain explanatory power to the tourism flow network. This indicates that, with the upgrading of consumption, tourists pay more and more attention to the quality of tourism products. Hence, the level of tourism service will directly affect tourists' perceptions of their destination image [42], and service quality reputation will have a direct impact on the tourism flow.

## 4. Conclusions

### 4.1. Findings

The inflow of tourists to a destination is affected by tourism resource endowment, transportation convenience, economic development levels, tourism reception, service capacity, and other factors. Based on the dual network perspective, this study uses SNA and GIS to analyze the tourism flow. It discusses the network characteristics, spatial structure, and node role; establishes an evaluation index system; and applies QAP to conduct correlation analysis and regression analysis on the influencing factors of tourism flow networks. This study's methodology is a bold attempt, that enriches the content of tourism flow research.

First, compared with studies that derived data entirely from tourist sample surveys and were purely based on social considerations [4,9], this study adopts the data published by tourists in their travel diaries with complete travel routes. Using these data, SNA can reflect well the relationship between the tourism flow network nodes (e.g., network density and centrality), while GIS analysis can better reflect the physical state of tourism flow networks, including their spatial structure and accessibility, so that the inferences are based on social and physical phenomena.

Second, this study analyzes the driving factors of the formation of a tourism flow network, and expands the network's research scope. At present, this mainly focuses on the density, centrality, and core degree of tourism flow network nodes [5]. The driving factors have an important impact on the pattern, formation and evolution of tourism flow networks, but there are few studies. Therefore, QAP analysis was carried out to measure the impact of various drivers on the nature, direction, and extent of tourism flows. The results show that tourism resource endowment, transportation convenience, economic development levels, tourism reception, and service capacity have significant impacts on tourism flow networks. In addition, the nature, direction, and extent of the impact of the various drivers are different.

In addition, although previous scholars have conducted extensive studies on tourism flows by using multi-disciplinary methods [5,16–19], these have been limited to conducting comprehensive and systematic studies on the network characteristics, spatial structure, and nodal functions of tourism flows from the dual networks perspective, using either SNA or GIS methods. A combined analysis through SNA and GIS can further examine the formation, evolution, and characteristics of tourism flow networks in urban agglomerations of the Yangtze River Delta [3,8], and draw more insightful conclusions from the relationship structure and spatial structure of network nodes. Strategically, this combination of SNA and GIS approaches can facilitate the assessment of tourism flow networks within geographical areas.

The empirical results of this study verify the applicability of SNA and GIS combined. Specifically, by analyzing the structure of the tourism flow network and measuring its density values in 2014, 2017, and 2020, the scale and evolution of the tourism flow network can be effectively judged [4,10]. Through the comparative analysis of the tourism flow network density in 2014, 2017, and 2020, it is found that the density of the tourism flow network shows an overall growth trend, and the spatial differences of the network density keep narrowing, indicating that the tourism flow network presents a perfect and balanced development trend. In addition, the overall network density is much lower than the average network density of the provinces under its jurisdiction, indicating that, influenced by administrative boundaries, the tourism flow frequency within provinces is significantly higher than that between provinces.

Moreover, by analyzing the nodal structure and measuring the degree centrality index of the nodes of tourism flow network in 2014, 2017, and 2020, the role, spatial difference, and evolution of network nodes can be effectively judged [48,49]. Through the comparative analysis of node centrality of tourism flow networks, it can be found that: ① The spatial difference of degree centrality gradually becomes balanced, and the tourism flow network evolves in a multi-core direction. Besides Shanghai, Nanjing, Suzhou, Yangzhou, and Hangzhou, the five core cities, Hefei, Wuxi, Huzhou, Ningbo, and other cities can be cultivated as the new core of the tourism flow network. ② On the whole, the difference of closeness centrality shows a decreasing trend, which indicates that the tourism flow of each node city increases, the tourism flow network structure has more point dependence, and more and more city nodes have high independence in the network. ③ The average value of betweenness centrality increases significantly, indicating that power in the tourism flow network is becoming more and more concentrated, and the core nodes in the network are gradually strengthening their control over other nodes. In addition to Shanghai, Nanjing, and Hangzhou, new urban nodes, such as Wuxi, Yangzhou, Hefei, and Jiaxing, play an important role of "bridge" and "intermediary" in the regional tourism flow network.

Furthermore, a comparison and analysis of the core–periphery structure of the Yangtze river delta urban agglomerations tourism flow network in 2014, 2017, and 2020 revealed the stability of the situation—the core of urban node degree value equalization—with the gap between the urban node degree high values at the core shrinking. However, the core degree of cities in low-value regions shows an obvious upward trend, and the number of cities in the core region keeps decreasing, indicating that the agglomeration effect of regional tourism flow is more and more significant.

Additionally, the formation of the tourism flow network is influenced by a variety of objective destination factors. The results of QAP correlation analysis show that tourism resource endowment, transportation convenience, economic development levels, tourism reception, and service capacity are positively correlated with tourism flow networks, which in turn are negatively correlated with the shortest travel time by automobile and high-speed rail/train, but positively correlated with the shortest travel time by air. QAP regression analysis shows that tourism resource endowment and tourism reception service capacity significantly promote tourism flow, while the minimum travel time of automobile and high-speed rail/train are significantly obstructed. Compared with previous studies, that focused on the single impact of traffic on tourism flow, this study believes that tourists tend to flow between cities with higher resource endowment and perfect tourism reception facilities. At the same time, cities with convenient transportation are more vulnerable to the spillover effects of tourism flows from central cities. Finally, owing to regional and industrial structure differences, and other reasons, government policies and economic development levels are considered to have no significant relationship with tourism flows. Therefore, the methods and results used in this study for driving tourism are applicable not only to the Yangtze River Delta region of China, but also to multiple destinations. However, this needs to be verified by other scholars in future empirical studies.

### 4.2. Implications, Limitations and Further Research

As a practical implication, the conclusion of research on the density of tourism flow networks, characteristics of nodes, spatial structure, and role of tourism flow networks is of great significance for developing regional tourism cooperation, planning tourism, designing tourism routes, and developing regional joint marketing. This effect is more obvious in the Yangtze River Delta region of China, as it has a high degree of regional tourism integration, and its tourism flow and tourists' spatial behavior are relatively complex. In this regard, this study is helpful to supplement and improve existing research, and provide new insights on tourists' spatial behavior preferences for government, corporate planners, and tourism service providers. On this basis, it is suggested that the government, corporate planners, and tourism service providers should first have a clear understanding of the structural characteristics of the regional tourism flow network before carrying out infrastructure

construction or marketing activities. It is very important to establish a tourist distribution center at the nodes of important tourist cities because it can provide tourists with enough information and services, which can help them effectively identify the route to their tourist destination. In addition, to avoid congestion in urban nodes, transport-related facilities and services cannot be ignored.

However, there were some limitations in this study. Although it innovatively combines SNA and GIS methods, its analysis is not enough to solve all the problems of tourism flow network formation. Especially due to the limitation of paper length, results based on these two methods do not provide sufficient details of social networks in the study area. In addition, no data or analysis method is perfect. As most of the people who are willing to publish travel diaries on OTA website belong to the youth group, although this study collected as many travel diary samples as possible, the travel behaviors reflected by them are basically the travel behaviors of the youth group. In other words, the route data obtained from the travel diaries cannot completely reveal the spatial movement rules of all tourists, and the robustness of the results is also affected by the data quality. Therefore, in order to better explain the driving mechanism of tourism flow network formation and evolution, future studies should strengthen the analysis of multiple stakeholders: government, enterprises, residents, tourism practitioners, and tourists on the one hand, while on the other hand, the data of tourist routes in questionnaires, internal materials of tourism enterprises, tourism newspapers, professional tourism market research institutions' annual reports, and tourist diaries can be summarized, so as to draw a more accurate conclusion that can reflect the formation of tourism flow networks. In addition, there are a wide range of factors that influence tourists' choice of destination. In addition to the factors listed in this study, natural ecological environments, entrepreneurs and other factors can also affect the formation of tourism flow network. For example, tourism entrepreneurs within different industries, society and culture as the core attraction, active group development to create a combination of point, line and surface of all kinds of tourism industry clusters, are also conducive to the formation of tourism flow. Future studies can consider incorporating more influential factors to obtain more accurate results on the formation of the tourism flow network.

**Author Contributions:** Y.W. contributed to all aspects of this work; H.C. wrote the main manuscript text; M.X. and C.L. analyzed the data. All authors have read and agreed to the published version of the manuscript.

**Funding:** This research study was supported by the National Social Science Foundation of China (Grant No. 19BGL145).

**Institutional Review Board Statement:** Not applicable.

**Informed Consent Statement:** Not applicable.

**Acknowledgments:** The authors would like to acknowledge all experts' contributions in the building of the model and the formulation of the strategies in this study. All individuals included in this section have consented to the acknowledgement.

**Conflicts of Interest:** The authors declare no conflict of interest.

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
