# Peer review of "Evolution and Driving Mechanism of Tourism Flow Networks in the Yangtze River Delta Urban Agglomeration Based on Social Network Analysis and Geographic Information System: A Double-Network Perspective"

_sustainability, doi:10.3390/su14137656_

Round 1

Reviewer 1 Report

Dear authors,

The abstract needs to be improved, Including motivation, objectives, theoretical framework, methodology, and main results, in a brief and concise manner. In the abstract, you should summarise the conclusions of your work, otherwise, it is more like the conclusion section. It does not appear that your work is new in the scientific literature. You should clarify what scientific gap your research covers. It should be coherently structured in a single paragraph.

The last sentence of the abstract adds nothing to a scientific article, and I recommend its deletion. “This study’s methodology is a bold attempt that enriches tourism flow research.”

In the introduction section, you can try to answer some points of interest. 1. a practical introduction answers three sets of questions: a. Who cares? What is the topic or research question, and why is it exciting and important in theory and practice? b. What do we know, what don't we know, and so what? What central, unaddressed puzzle, controversy, or paradox does this study address, and why does it need to be addressed? c. What will we learn? How does your study fundamentally change, challenge, or advance scholars' understanding?

In the introduction, you include part of the theoretical framework. This is the weakest part of your article, and you should consist after the introduction, a section where you have the conceptual works on tourism flows and the problems developed in the last two decades, as well as their relationship with the interest in sustainability, highlighting over-tourism, the seasonality of tourism flows and the role of technological innovation, for this, I recommend the reading and inclusion of the following works:

Puertas Medina, R. M., Martín Martín, J. M., Guaita Martínez, J. M., & Serdeira Azevedo, P. (2022). Analysis of the role of innovation and efficiency in coastal destinations affected by tourism seasonality. Journal of Innovation & Knowledge, 7(1), 100163.

Martín, J. M. M., & Fernández, J. A. S. (2022). The effects of technological improvements in the train network on tourism sustainability. An approach focused on seasonality. Sustainable Technology and Entrepreneurship, 1(1), 100005. https://doi.org/10.1016/j.stae.2022.100005

Theoretical framework

In your study, you do not try to develop a theoretical background, and your study’s conceptual model is not discussed. You do not have a theoretical framework that is related to helping improve tourism. It would help if you searched the very good literature of developing new procedures for tourism promotion so that you can find a suitable model related to your findings and then develop and discuss the propositions of your research model.

It should also raise the status quo of the situation: Why should the economic impact of tourism be promoted and analyzed?

The increase in overtourism in several places of the world must appear in your introduction, I suggest you include this point of view in your study supported by recent scientific literature,

Also, in relation to sustainable tourism, this idea needs to be further supported by the scientific literature, remember, this is a “Sustainability” journal.

At the end of the conclusions, I recommend you introduce the role of the entrepreneur. Regarding the methodology, many indices measure the most relevant factors influencing tourism competitiveness, and you should name these in your study.

The global and national economic situation is also relevant for boosting tourism.

  • Martínez, J. M. G., Martín, J. M. M., & Rey, M. D. S. O. (2020). An analysis of the changes in the seasonal patterns of tourist behavior during a process of economic recovery. Technological Forecasting and Social Change161, 120280.

Methodology

It will be necessary to justify why this methodology was selected and not others.

Conclusion:

As your findings are interesting, you can suggest a conceptual model based on them. This conceptual model can be tested in future research. However, you can at least discuss and develop some propositions based on the relationships of your conceptual model. This section will need to be expanded with the limitations of this work and future lines of research with more details that you have presented. Finally, it could be interesting to highlight the practical significance for organizational members of this study and reference policy prescriptions derived from this analysis, as well as the implications for future research. I will encourage the authors to expand the agenda for future research.

Reviewer 2 Report

1. General Comments

I have the great interest in reading this article. After a thorough reading, I feel a little bit confused by the current version. I am failed to find the uniqueness, so indicated as a joint use of SNA and QAP,  highlighted by the authors. This kind of application is not really an innovation in this field. Furthermore, the results based on these two methods did not provide enough details of the social network in the study area. The robustness of the results is affected by the data quality, which is not guaranteed  in this study.

2. Specific comments

1) Introduction

I also feel confused by the research motivation. What we can learn from the research findings, of which most are intuitive results? I will suggest a big improvement of the introduction to clarify why the authors need and could conduct this study. 

2) Analytical framework

The article badly needs an analytical framework or a conceptual analytical framework, which can well indicate what kind of causal relationships to be investigated by this study. Please note most readers and scholars are interested in casual relationships rather than co-relationship. 

The authors also need to note the difference of methods (narrowly defined in this study) and theories. The methods, mainly quantitative methods, are indicated with large space (possibly no need occupy these space). However, a few indication of theory. This is  a significant weakness of this study.

3) Model

For the current regression model, it must need a statistical description of all variables used in the model. Right now, we have no idea about what is the situation of dependent variable in the model. 

4)Data

The dataset obtained from two websites must need  be clarified. What are the variables of the dataset? and What are the measurement methods of the variables? What is the size of the dataset?

5)Results

We find some slight changes of the temporal results (Figure 1 - 4)? Why? What are the more practical results of tourism flow at the sample cities level?  

6) Grammar

Though I am not an English native speaker, I will say that the language could be improved to some extent. For, format of section 2.1 needs not italicized.

Reviewer 3 Report

The very interesting research approach and study results have been presented in the paper. Purpose of research was to identify spatial features and to consctruct a tourism flow network for the big tourist region the Yangtze River Delta Urban Aglomeration - testing much innovative methodical approach and support / give better basis the government policy makers and tourism development decision makers.

The important value of the paper is the largely innovated methodological approach consisting on using combined method: SNA (Social Network Analysis) and GIS methods - for needs of tourist flow spatial network. SNA-GIS methods have been supported by QAP (Quadratic Assigment Procedure).

As research results show, this methodical approach better helped to  recognise and even better undestand the spatial behaviour and flow of tourists.

The study area is also very interesting: The Yangtze River Delta urban area - a huge economic and tourism region, with very rich tourism resources including tourism system. The results of the used innovative methodical approach are better visible in a such big region, with 26 cities within its borders (the identified network of tourist flow is clearly visible in a big area).

Section 2. Materials and Methods. The content is presented synthetically and clearly. My suggestion is to add the Figure with the location of the Yangtze Delta study area in China. Section 2.1 - the content is written in Italics - it should be changed into a normal style.

Section 3. Results and Discussion. The content is written well, with good references to literature -  very interesting for a reader.

Section 4. Conclusions. Written well. As mentioned above, the largely innovative methodological approch is the main (oryginal) value of the paper. So, the Authors should emphasize advantages and disadvantages/difficulties of this methodical approach. And the Authors did it in Sections 3 and 4. My suggestion is: try to do it more distinctly. 

In general. Very good, oryginlal and interesting paper. The most valuable  element: much innovative methodical approach - enriching the general methodical arsenal in this regard (research concerning spatial structure of tourist flow in big areas). 

Round 2

Reviewer 1 Report

Congratulations! The authors have effected all suggestions proposed by reviewers in order to improve your study. 

Reviewer 2 Report

The authors  have followed the comments and finished the required revisions.  I have no more suggestions.